# Fabricated AIE-Based Probe to Detect the Resistance to Anoikis of Cancer Cells Detached from Tumor Tissue

**DOI:** 10.3390/cells11213478

**Published:** 2022-11-03

**Authors:** Ya-Nan Chang, Yuelan Liang, Jiayi Wang, Ziteng Chen, Ruyu Yan, Kui Chen, Juan Li, Jiacheng Li, Haojun Liang, Gengmei Xing

**Affiliations:** 1CAS Key Laboratory for Biomedical Effects of Nanomaterials & Nanosafety, Institute of High Energy Physics, Chinese Academy of Sciences, Beijing 100049, China; 2Institute of Physical Science and Information Technology, Anhui University, Hefei 230601, China; 3Beijing Synchrotron Radiation Facility, Institute of High Energy Physics, Chinese Academy of Sciences, Beijing 100049, China

**Keywords:** anoikis, aggregation-induced emission, probe, fluorescence, cancer cell

## Abstract

(1) Background: Resisting anoikis is a vital and necessary characteristic of malignant cancer cells, but there is no existing quantification method. Herein, a sensitive probe for assessing anoikis resistance of cancer cells detached from the extracellular matrix was developed based on the aggregation-induced emission (AIE) of AIEgens. It has been reported that detached cancer cell endocytose activated integrin clusters, and in the endosome these clusters recruit and activate phosphorylate focal adhesion kinase (pFAK) in the cytoplasm to induce signaling that supports the growth of detached cancer cells. (2) Methods: We established a lost nest cell model of cancer cells and determined their ability to resist anoikis. The colocalization of the activated integrin, pFAK, and endosomes in model cells was observed and calculated. (3) Results: The fluorescence signal intensity of the probe was significantly higher than that of the integrin antibody in the model cells and the fluorescence signal of probe signal was better overlapped with labeled pFAK by fluorescence in endosomes in model cells. (4) Conclusions: We developed a quantitative multi-parametric image analysis program to calculate fluorescent intensity of the probe and antibodies against pFAK and Rab5 in the areas of colocalization. A positive correlation of fluorescence signal intensity between the probe and pFAK on the endosome was observed. Therefore, the probe was used to quantitatively evaluate resisting anoikis of different cancer cell lines under the lost nest condition.

## 1. Introduction

Metastatic colonies established by cancer cells that detach from primary tumors are responsible for the vast majority of cancer deaths [1]. There is increasing effort to label these detached cancer cells and evaluate their malignancy for detecting the curative effects of cancer in clinic and selecting novel therapies to block tumor metastasis.

Normal epithelial cells require attachment to the extracellular matrix (ECM) to survive. If the attachment is lost, these cells undergo an anchorage-dependent apoptosis termed anoikis [2,3]. Cancer cells that detach from the ECM can adopt a distinct cellular morphology and molecular features to facilitate long-term survival in the circulatory system [3,4]. These cancer cells from the primary focus migrate into the distal tissue to grow and develop forming new metastatic clones, and lead to patients’ death. One attractive idea is to design probes that can detect the anoikis resistance of these detached and freed cancer cells. The probe could provide an observable index to evaluate tumor malignancy grade and promote the development of therapeutics. In addition, such a probe would enable a better understanding of what enabled the survival of detached cancer cells during the metastatic cascade. The ability of ECM-detached cancer cells to evade anoikis has been researched for many years. Molecules such as epidermal growth factor receptor (EGFR) [5], transforming growth factor (TGF) ß-activated kinase [6], and small GTPases can facilitate their survival during the metastatic cascade of detached cancer cells [3,7,8]. ECM–integrin signaling regulates various cellular functions such as cell survival and migration [9], and the anoikis resistance of detached cancer cells [10]. In the process of detaching from the ECM, activated integrin clusters at the site of cancer cell–ECM adhesions are endocytosed into endosomes, and then recruit focal adhesion kinase (FAK) in the cytoplasm to localize on endosomes, producing an endosomal platform to support the survival of cancer cells after integrin signaling away from the plasma membrane [11] and to facilitate their metastasis. Activated integrin clusters in the endosome are required for interactions with FAK to serve as integrin-mediated signaling platforms [12]. Building such platforms for integrin-mediated FAK signaling is, therefore, strongly dependent on activated integrin clustering in the endosome. Clustered integrin in the endosome of detached cancer cells would be used as a signal to investigate how these cells resist anoikis.

Fluorescent materials such as semiconducting quantum dots and organic dyes have been used to label biological molecules to investigate the alteration of their function in physiological and pathological processes [13,14]. Although conventional fluorescence probes have high sensitivity, their aggregation-caused quenching (ACQ) blocks the detection of the integrin cluster in the endosome in a confined space. A special fluorescent material called aggregation-induced emission luminogen (AIEgen) was reported by Tang’s group in 2001 [15] and has been used in several studies [16,17,18,19,20,21,22]. AIEgen can overcome the ACQ phenomenon to allow labeling of endosomal integrin clusters.

In this study, we designed and structured an AIE probe, and evaluated its selectivity, feasibility, and optimal concentration. A quantitative multi-parametric image analysis program was developed to determine the capacity of cancer cells to resist anoikis based on detecting activated integrin clusters that mediated endosomal FAK signaling to support the survival of detached cancer cells.

## 2. Materials and Methods 

### 2.1. Synthesis Scheme and Characterization of AIE–cRGD

4,4-(1,2-Diphenylethene-1,2-diyl) dibenzoic acid (AIE, Sigma, Merck KGaA, Darmstadt, Germany) (2.5 mg, 6 μmol) was dissolved in DMSO (0.5 mL, Sigma, Merck KGaA, Darmstadt, Germany), then a catalytic amount of EDC (50 μL) and NHS (30 μL) was injected into the AIE solution. The mixture was reacted for 30 min by ultrasound. Subsequently, a solution of c (ARGD-3-amino benzoic acid) (cRGD, Scilight Biotechnology, Beijing, China) (3.2 mg, 6 μmol) in 1 × PBS buffer (0.64 mL) was added to the above solution to further reaction overnight at room temperature. The expected product was named AIE–cRGD. Then, product AIE–cRGD was characterized by capillary electrophoresis.

### 2.2. Spectral Property of AIE–cRGD

The UV absorption of AIE, AIE–cRGD was measured by double beam ultraviolet spectrophotometer (Persee, Beijing, China) from 200 to 450 nm. The AIE, AIE–cRGD was diluted in various concentrations in DMSO/water (*v*/*v* = 1/100). The fluorescence intensity of AIE–cRGD was measured by fluorescence spectrophotometer (Shimadzu, Kyoto, Japan) under an excitation of 360 nm. The AIE–cRGD was dissolved in 1×PBS to obtain a series of concentrations.

### 2.3. The Selectivity Response of AIE–cRGD

The recombinant integrin protein (R&D systems, Minneapolis, MN, USA) was dissolved in water, diluting to obtain the integrin solution (100 μg/mL). The 1 mg bovine serum albumin (BSA) (Amresco, Houston, TX, USA) was dissolved in 1 mL PBS, and diluted to 100 μg/mL. Then, 5 μL AIE-cRGD or (and) BSA was added to 300 μL PBS, then UV absorption intensity was measured by double beam ultraviolet at 230 nm.

### 2.4. Cell Culture

The cancer cell lines (MCF-7, MDA-MB-23 (human breast cancer), Hela (human cervical carcinoma), HepG2 (human liver cancer), A549 (human lung cancer), Caco-2 (human colon carcinoma), DU-145 (human prostate cancer), U87-MG (human glioblastoma), and KB (human Carcinoma)) were provided by China Infrastructure of Cell Line Resource (Beijing, China). MCF-7, Hela, HepG2, KB, and 143B cells were cultured in DMEM–high glucose (Hyclone, Logan, UT, USA) containing 10% fetal bovine serum (FBS, PAN-Biotech GmbH, Adenbach, Germany), 1% penicillin streptomycin (Gibco, Thermo Fisher Scientific, Waltham, MA, USA) at 37 °C under 5% CO_2_, and MDA-MB-231 cells were cultured in L15 medium (Boster, Wuhan, China) containing 10% FBS at 37 °C without CO_2_. A549, DU-145, and U87-MG cells were cultured in RPMI 1640 medium (Hyclone, Logan, UT, USA) supplemented with 10% FBS at 37 °C under 5% CO_2_, Caco-2 cells were cultured in minimal essential medium (MEM, Gibco) containing glucose and 1% non-essential amino acids (NEAA, Gibco), 50 U/mL penicillin and 50 μg/mL streptomycin, 10 mM HEPES (Sigma, Merck KGaA, Darmstadt, Germany), and 20% FBS at 37 °C under 5% CO_2_.

### 2.5. Cytotoxicity Assay

The cancer cells (MCF-7, MDA-MB-231) were resuspended, and 9 × 10^3^ cells were seeded into each well of a 96-well plate. After 24 h, a series of concentrations of AIE–cRGD (1, 10, 50, 100, 500 μg/mL) were added to treat the cells for 24 h. Commercial Cell Counting-8 (CCK-8, Dojindo, Kyushu, Japan) was used to assess the cell viability.

### 2.6. ITC

ITC data for the titration of AIE probe (100 μM) with protein integrin α_v_β_3_ (5 μM). The integrin injections were 2 μL, spaced at 3 min intervals. There was an initial endothermic peak after each injection, which corresponds to heat of dilution. The thermal baseline then stabilized to a lower power level, due to exothermic catalysis of integrin α_v_β_3_ by AIE probe. The power generated at each substrate concentration was proportional to the rate of reaction.

### 2.7. Cells Imaging

MDA-MB-231 cells were pre-cultured in confocal Petri dish. The cells were washed with PBS, fixed with 4% paraformaldehyde for 20 min, and treated with 0.1% Triton X-100 for 20 min. Further, the cells were continued to incubate AIE–cRGD or AIE at a series of different concentrations (1, 5, 10, 50, 100 µg/mL, in PBS) overnight at 4 °C, respectively. Then, imaged by laser scanning confocal microscope (LSCM, Nikon A1, Tokyo, Japan).

### 2.8. Colocalization with Integrin-Labeled Antibody

MCF-7, MDA-MB-231 cells were pre-cultured in confocal Petri dish. The cells were fixed with 4% paraformaldehyde and incubated in blocking buffer (5% BSA and 0.3% Tween-20 in PBS) for 2 h at room temperature. A solution of AIE–cRGD at a concentration of 100 µg/mL was added to cells to continue to incubate overnight at 4 °C. After then, the cells were washed with PBS, incubated with primary antibody of integrin α_v_β_3_ (Abcam, Cambridge, UK) overnight at 4 °C and further incubated with the second antibody (Abcam, Cambridge, UK) for 2 h at room temperature. Stained cells were observed by LSCM.

### 2.9. Distribution of Integrin in Endosomes

MCF-7, MDA-MB-231 cells were pre-cultured in confocal Petri dish. The AIE–cRGD solution (100 µg/mL) was added to incubate for 2 h at 37 °C, respectively. After that, the cells were washed with PBS, fixed with 4% paraformaldehyde and incubated in blocking buffer for 2 h at room temperature, then incubated with primary antibody of Rab5 (Abcam) and Rab7 (Abcam) overnight at 4 °C and incubated with the second antibody (Alexa Fluor 594/488, Abcam) for 2 h at room temperature, respectively. Stained cells were observed by LSCM.

### 2.10. Construction of Anoikis-Resistance Model

The Annexin V-FITC/PI apoptosis detection kits (Dojindo, Kyushu, Japan) was used to detect apoptosis by flow cytometer (FCM, Accuri C6 BD, Piscataway, NJ, USA) as per manufacturer’s instructions. The 6-well plates were coated with 30% poly (2-hydroxyethyl methacrylate) (poly-HEMA, Sigma, Merck KGaA, Darmstadt, Germany) dissolved in 96% ethanol solution per well and letting the wells dry out overnight. The cell suspensions (3 × 10^5^ cells/well) of MCF-7 and MDA-MB-231 cells were seeded into poly-HEMA-coated 6-well plates for 24 h, 48 h, and 72 h, respectively. Afterwards, the cells were collected and stained as per the Annexin V-FITC/PI apoptosis detection kits manufacturer’s instructions. The stained cells were detected by FCM.

### 2.11. Fluorescent Dye of Anoikis-Resistant Cells

In this study, 3 × 10^5^ cells/well of MDA-MB-231 cells or the other cancer cells (Hela, HepG2, A549, Caco-2, DU-145, U87-MG, KB cells) were cultured in poly-HEMA-coated 6-well plates. After incubating 24 h, the cells were collected, fixed with 4% paraformaldehyde and incubated in blocking buffer for 2 h, then the cells were incubated with AIE–cRGD (100 μg/mL) or antibody of integrin αvβ3, then incubated with primary antibody of pFAK (Life Technology, Thermo Fisher Scientific, Waltham, MA, USA) and Alexa Fluor 594 (Abcam, Cambridge, UK) for 2 h at room temperature, and following that, incubated with anti-Rab5 antibody (BD Biosciences, Franklin Lakes, NJ, USA) and second antibody with Alexa Fluor 488 (Abcam, Cambridge, UK) for 2 h at room temperature, respectively. The stained cells were imaged by LSCM.

## 3. Results and Discussion

### 3.1. Probe Synthesis and Characterization

To generate the AIE-cyclic arginine-glycine-aspartic-conjugated (cRGD) probe (Figure 1A), the carboxyl of an AIEgen molecule (4, 4-(1, 2-Diphenylethene-1,2-diyl) dibenzoic acid) and amino of cRGD were combined via amidation. Because the AIEgen contains two para-carboxyls, the reaction can be altered with varied ratios of AIEgen to cRGD (1:1 or 1:2). We investigated the optical properties of AIEgen with one or two cRGD. The UV absorption peak of the probe was still observed at 230 nm as AIEgen, and the absorption intensity was maintained (Figure 1B). The emission spectra also showed that the fluorescence property of this probe was not significantly altered by linking cRGD (Appendix A). Therefore, we think that the cRGD ratio modified on AIEgen will not cause changes in the UV absorption spectrum, and the 1:1 method is selected for probe synthesis. For the AIE–cRGD, cRGD decreased the charge of AIEgen, and electrophoretic migration speed was between AIE and cRGD. [21] The results also confirm that cRGD had linked with AIEgen to form AIE–cRGD (Figure 1C).

### 3.2. Evaluating the Selectivity of the AIE–cRGD to Label Specific Proteins

We further investigated probe selectivity by adding various biomolecules into the solution. We hypothesized that the specific binding between AIE–cRGD and integrin α_v_β_3_ would restrict the intramolecular rotation of the aromatic rotors of the AIEgen to induce excitation of optical properties; conversely, nonspecific binding did not activate the optical signal. Therefore, we measured the fluorescence spectrum in 365 nm to assess its selectivity (Figure 1D). Bovine serum albumin (BSA), matrix metalloproteinase 9 (MMP9), pepsin, trypsin, glutamine, cystine, cysteine, or human integrin α_v_β_3_ proteins were added to the test system (Mole ratio was 1:1), respectively. As can be seen from the results of the fluorescence spectrum, the fluorescence intensity of the integrin α_v_β_3_ group was significantly higher than that of the control group and other groups, which proved that the probe had specific binding with integrin α_v_β_3_. Meanwhile, we quantified the ultraviolet absorption intensity of the probe to detect the electron transitions intramolecular and between the molecules. As shown in Figure 1E, among the tested proteins, integrin α_v_β_3_ caused a significant ~115-fold increase in the ultraviolet absorption intensity of the AIE–cRGD probe, from 0.0027 to 0.31. BSA, MMP9, pepsin, trypsin, glutamine, cysteine, and cysteine did not have this effect. It is suggested that the specific binding can reduce intramolecular rotation and vibration, and increase intramolecular and intermolecular electronic transition, with more energy released by optical properties. These results suggested that the probe had excellent selectivity for integrin α_v_β_3_. 

### 3.3. Assessing the Ability of the Probe to Label Activated Integrin in Cells

Integrin is an important transmembrane adhesion receptor protein [23,24]. Upon receiving an external signal, the integrin on the cell surface rapidly changes from a resting state with low binding to a tense state with high binding activity [25], and the activated integrin forms clusters [26]. The probe designed here is a fluorescent probe based on the aggregation induced luminescence phenomenon. Therefore, it can be speculated that when the probe is bound to free and dispersed integrin, aggregation occurs less, and fluorescence is weak. On the other hand, if the protein is in an aggregation state, the probe will emit light after binding to it, which can also prove the aggregation state of integrin. In electrophoresis experiments, the purified and free integrin α_v_β_3_ were used; we incubated the one group of protein bands with integrin α_v_β_3_-specific antibodies, and then with HRP secondary antibody; the other group was incubated with the AIE-cRGD, and then the fluorescent bands were detected under UV irradiation. The results showed that the protein bands could bind to the antibody and emit fluorescence, while the bands bound to the probe could not be detected (Figure 2A). This is because in the process of electrophoresis and membrane transfer, the integrin monomer cannot aggregate, so the aggregation state and fluorescence are weak after binding with the probe, and the protein cannot be detected. In ITC experiment, we used the probe to titrate the integrin. After measurement and calculation, the binding ratio of the two was 6.59 ± 1.11:1 and the binding constant K was 1.22 E5 ± 9.79 E4, which proved that the two had a strong binding effect (Figure 2B). Meanwhile, the probe was used to label the activated integrin clusters in MCF-7 and MDA-MB-231 cells. A fluorescent antibody was also used to label the integrin to evaluate the targeting ability of the probe in cells. Laser scanning confocal microscope (LSCM) images (Figure 2C) showed that the probe (blue) was highly colocalized (cyan) with the fluorescent antibody (green) in both MCF-7 and MDA-MB-231 cells. The corresponding colocalization coefficients in these cell types were 0.954 and 0.837, respectively (Figure 2C), and the intracellular fluorescence signal of the AIE-cRGD probe was stronger than that of the antibody. This indicated that the binding with activated integrin clusters induced probe aggregation, which restricts the intramolecular rotations of the aromatic rotors in AIEgen and induces the fluorescence signal. Conversely, the aggregation effect could cause the fluorescence signal of the antibody to be weakened by ACQ [27], so the fluorescence signal of activated integrin labeled by the probe was significantly stronger than the fluorescent antibody label. In addition, the lower rate of colocalization in MDA-MB-231 cells (0.837) than in MCF-7 cells (0.954) also indicates that MDA-MB-231 cells have higher viability than MCF-7 cells because they contain more activated integrin clusters to mediate cell growth signals. Collectively, these findings suggest that the AIE probe could be used to target the integrin clusters and induce intracellular fluorescence emission.

### 3.4. Detecting the Suitable Probe Concentration for Labeling the Living Cells

Using CCK-8 kits to detect the cytotoxicity of the AIE probe, we found that concentrations ranging from 1 to 500 μg/mL did not obviously reduce MCF-7 or MDA-MB-231 viability at 24 h as compared with untreated cells. We further investigated the labeling efficacy of the probe on MDA-MB-231 cells at various concentrations (1, 5, 10, 50, 100 μg/mL). As the concentration of the probe increased, the fluorescent signal intensity in labeled cells was enhanced (Figure 3A). Using the probe which contain 50, 100 μg/mL AIEgen molecular and the same concentration of free AIEgen molecules, respectively, to label cells, the fluorescence signal of the probe in cells was obviously stronger than that of free AIEgen (Figure 3B). This suggested that the probe specifically bound integrin clusters, which restricted the intramolecular rotations of the aromatic rotors of the AIE molecule. The statistical data further confirmed that the fluorescence signal in cells labeled with 100 μg/mL probes was stronger than those labeled with 50 μg/mL probe (Figure 3C). Therefore, we chose a working concentration of 100 μg/mL for further experiments. 

### 3.5. Detecting Clustered Integrin Distribution in Cells with the Probe

Employing the poly-HEMA-induced method [28], we simulated cancer cells that had detached from the ECM and acquired the ability to resist anoikis as the model cells. The control cells were cultured in low adhesion plates for 24, 48, or 72 h, and then we detected their ability to resist anoikis by flow cytometry. The results show that the model MDA-MB-231 cells had a low apoptosis rate compared with control cells. At 24 h, the apoptosis rates of the control and model groups were 49.0% and 30.0%, respectively; at 48 h they were 57.6% and 48.7%; and at 72 h, apoptosis rates were similar at 65.4% and 63.6% (Figure 4). These values indicated that the model cells could resist anoikis for 24 h. Although the apoptosis rate of MCF-7 model cells was lower than the control cells, there was no significant difference. Then, we collected model cells from poly-HEMA plates for reculture on normal plates. After 2 h, the cellular morphology changed from gathered spheres in suspension to fusiform and began to attach and grow on the bottoms of culture plates (Appendix A). This verified that the model cells re-colonized and grew by adhering to the culture plate. Collectively, our results showed that model cells could successfully resist anoikis, so they were used in subsequent experiments.

Using the probe, we labeled integrin clusters in cancer cells (control or model) to detect their distribution. The cells were simultaneously labeled with a fluorescent antibody for Rab5 (a protein marker of early endosomes) and Rab7 (a protein marker of late endosomes), and colocalization of the probes and fluorescent antibodies was assessed to determine the intracellular location of the probe (Figure 5A). Fluorescence microscopy images (Figure 5B,C) showed that in MCF-7 or MDA-MB-231 model cells, the colocalization rate of the probe was higher with Rab5 than Rab7. This suggested that clustered integrin was most distributed in the early endosomes of these detached cancer cells. This was consistent with the report by Alanko and colleagues [10] that detached cancer cells build integrin-mediated signal platforms in early endosomes to support their survival.

### 3.6. Assaying the Establishment of Signal Platform in Model Cells of Resisting Anoikis

Detached cancer cells reportedly endocytose activated integrin into endosomes to establish signaling platforms to maintain viability and evade anoikis [10,11]. We used the probe in conjunction with fluorescent antibodies against pFAK and Rab5 to assess the endosomal integrin-mediated signaling.

Confocal images of labeled cells showed colocalization of fluorescence signal from the probe (blue) and antibodies to pFAK (red) and Rab5 (green) and indicated the integrin cluster recruiting pFAK on the endosome (Figure 6A). Colocalization was significantly higher in model cells compared to control cells (Figure 6B). To clearly observe triple labeling, we accurately detected colocalization in the endosomes of model and control cells using 3D LSCM images. With *z*-axis imaging, the probe and pFAK labels simultaneously appeared in endosome (white arrows in Figure 6) to a greater degree in model cells than in control cells (Figure 6C), and statistical analysis confirmed that the colocalization rates were significantly difference. These experiments were repeated in several cell types (U87, KB, HeLa, HepG2, DU-145, and Caco-2; Appendix A) with the same results. Although the colocalization rates varied because they differ from cell types, the higher rates were observed in model cells compared with control cells. These results demonstrate that the probe could be used to confirm the presence of activated integrin clusters in endosome-founded signaling platforms that allow detached cancer cells to resist anoikis for survival.

We also compared the efficiencies of the probe and integrin antibody in detecting colocalization of integrin clusters and pFAK. Compared with the fluorescent integrin antibody, the probe increased the colocalization rate of integrin clusters and pFAK by ~1.2 fold. (Appendix A). This indicated that the probe was more sensitive than the fluorescent antibody for detecting active integrin clusters.

### 3.7. Quantitative Assessment of Anoikis Resistance of Free Cancer Cells Based on Probe Fluorescence Parameters

Using triple-color automated confocal microscopy, we obtained images showing intracellular co-localization of pFAK, Rab5, and activated integrin. This indicated that integrin-mediated signaling platforms have been established on endosomes. Therefore, we developed a quantitative multi-parametric image analysis program to calculate fluorescent intensities of the probe in areas with colocalization.

First, we labeled model cells with the probe and fluorescent antibodies for pFAK and Rab5 and obtained confocal microscopy images (Figure 7A). We collected fluorescent signals in the area of 512 × 512 pixels and then extracted, measured, and calculated the fluorescence intensities of the three colors (red, blue, and green) in each pixel of the fixed area using the MATLAB program. We selected pixels with fluorescence intensities >0 for all three colors and named effective pixels. The results indicate that the internalized and activated integrin clusters in endosome recruit pFAK to build endosomal signal platforms. We further selected these aggregated colonies of effective pixels (white frame in Figure 7A) and drew a 3D image according to the varied fluorescence intensities of the three colors at every effective pixel (Figure 7B). In the 512 × 512 pixels area, the aggregated colony of effective pixels is surrounded by a large number of invalid pixels without triple colocalization. Secondly, the number of all effective pixels was counted (N_0_), and different color (red, green, and blue) pixels in the total area were respectively extracted and calculated (denoted N_R_, N_G_, and N_B_). N_0_ is the number of effective colocalization pixels, and N_0_/N_R_, N_0_/N_G_, and N_0_/N_B_ are indexes of the co-localization rates of every color. Compared with the control group, the colocalization rate of each color in the total area was obviously higher in the model cells (Figure 7C).

To develop a strategy to detect the capacity of detached cancer cells to resist anoikis, we analyzed the correlations between probe fluorescence intensity and pFAK antibody signal in areas with triple colocalization. We extracted the fluorescence intensities of pFAK and the probe in all effective pixels and calculated the correlation between them. We collected and statistically analyzed a total of 22 different cells images and calculated and analyzed the adjacent effective pixel points as colocation islands. We took the fluorescence intensity values of the probe as the horizontal coordinates and pFAK as the vertical coordinates and plotted all effective pixels to generate an intensity correlation curve (Figure 6D). After fitting data in the plot, a linear equation was obtained: *N*_pFAK_*=* 125.86 *+* 0.346 *N*_AIE-cRGD_ − 2.99 × 10^−5^
*N*_AIE-cRGD_(1)

*N*_pFAK_ means expression quantity of pFAK in endosomes, and *N*_AIE-cRGD_ means probe fluorescence intensity.

There was a significant positive correlation between probe fluorescence intensity and pFAK in endosomes. The probe signal intensity was negatively correlated with anoikis. This indicates that the fluorescence signal of the probe can be used to quantify endosomal signal, where pFAK is recruited to establish integrin-mediated signaling to support the survival of cancer cells following ECM detachment. Importantly, the results suggest that the probe we designed could be useful for evaluating the cellular capacity of resisting anoikis.

## 4. Conclusions

We fabricated an AIE–cRGD probe that specifically binds to integrin αvβ3 and exhibits high fluorescence emission because the activated integrin cluster restricts intramolecular rotation. In model cells, the probe label clearly colocalized with endosomal pFAK, suggesting the activated integrin cluster recruited pFAK to build endosomal signal platforms to resist anoikis for survival. The probe was also more sensitive than the fluorescent antibody for detecting active integrin. To more accurately evaluate resisting anoikis using the probe, we developed a quantitative multi-parametric image analysis program to achieve more accurate tricolor colocalization. Statistical analysis revealed a significant positive correlation between probe fluorescence intensity and endosomal pFAK. Similar detection results were obtained in a variety of tumor cells. This analysis method reads and counts every pixel in the whole image, so the image data can be displayed in data, which is clearer and more intuitive. This demonstrates that the probe’s fluorescence signal can be used to quantify the signal platform on the endosome using a correlation equation and the probe could be used to detect anoikis resistance in detached cancer cells. Thus, a novel probe for evaluating the degree of cancer cell malignancy by detecting their ability to resist anoikis was fabricated. However, the principle of this probe is aggregation and luminescence, the detection depends on the dissolution effect of the probe itself. Reducing background noise of the probe itself is the future direction of improvement.

## Figures and Tables

**Figure 1 cells-11-03478-f001:**
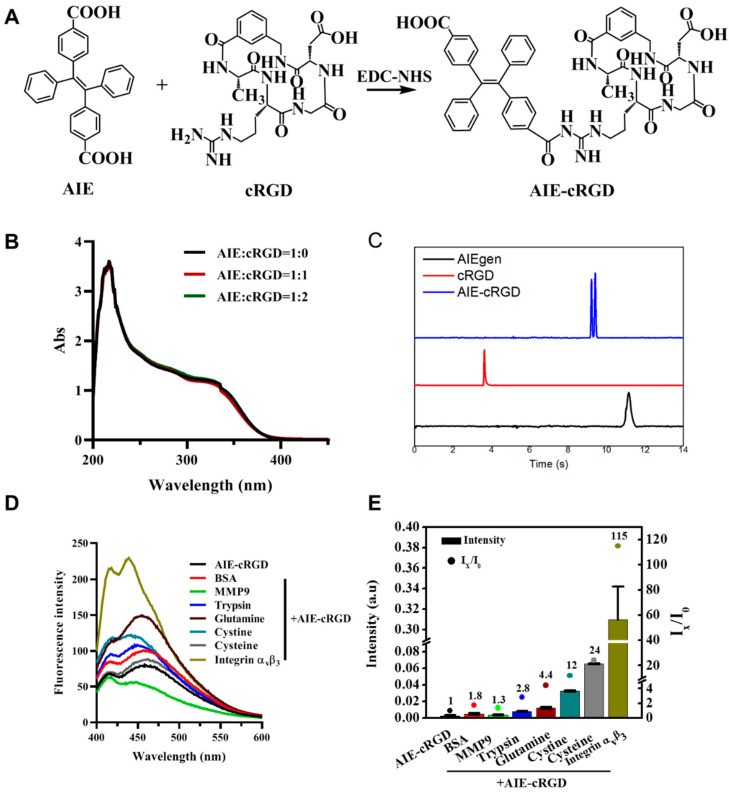
Synthesis and characterization of AIE–cRGD. (**A**) Synthesis scheme of AIE–cRGD. (**B**) UV spectra of different proportion of AIE with cRGD. AIE/cRGD = 1/0 (blank line), AIE/cRGD = 1/1 (red line), AIE/cRGD = 1/2 (green line), respectively. (**C**) Capillary electrophoresis characterization of AIE–cRGD, the blue line means AIE–cRGD, the red line means cRGD, and the black means AIE molecular. (**D**) The fluorescence intensity of AIE–cRGD, when Ex= 365 nm with various target biomolecules (n (AIE–cRGD)/n (biomolecules) = 1:1). (**E**) The intensity of ultraviolet absorption of AIE–cRGD at 230 nm with various target biomolecules.

**Figure 2 cells-11-03478-f002:**
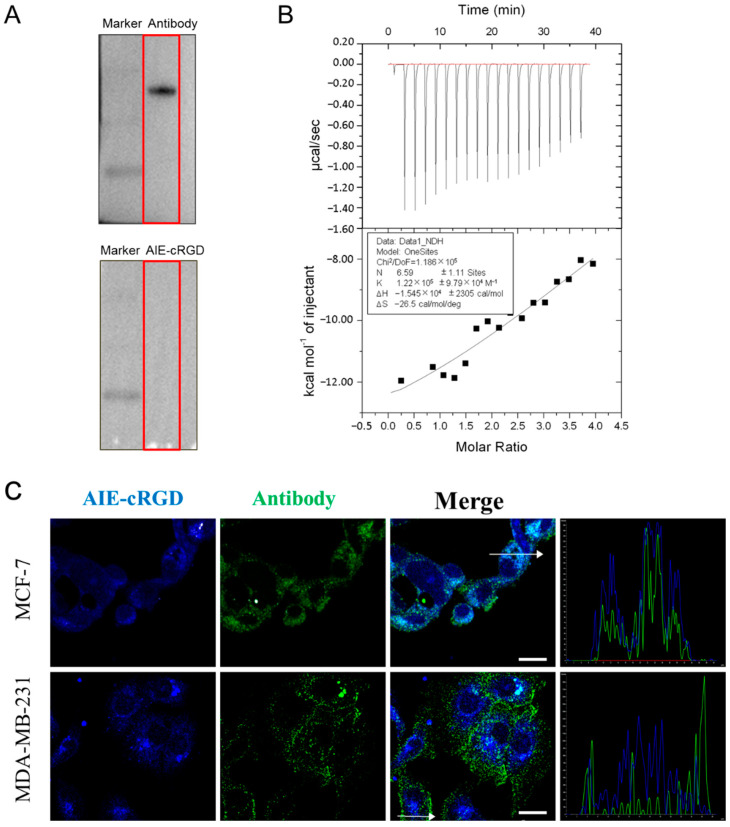
(**A**) Western blotting test for integrin α_v_β_3_ proteins. (**B**) The ITC data for measurement of reaction rate of AIE-cRGD (100 μM) with protein integrin α_v_β_3_ (5 μM). (**C**) Colocalization coefficients of AIE–cRGD and anti-integrin antibody in MCF-7, MDA-MB-231. Bar = 20 µm. The cells were incubated with AIE–cRGD (100 μg/mL), followed by further incubation with anti-integrin α_v_β_3_ antibody overnight at 4 °C. The curve graph is a result of measuring the intensity of two kinds of fluorescence at random position of white arrows in the merge diagram of two cells.

**Figure 3 cells-11-03478-f003:**
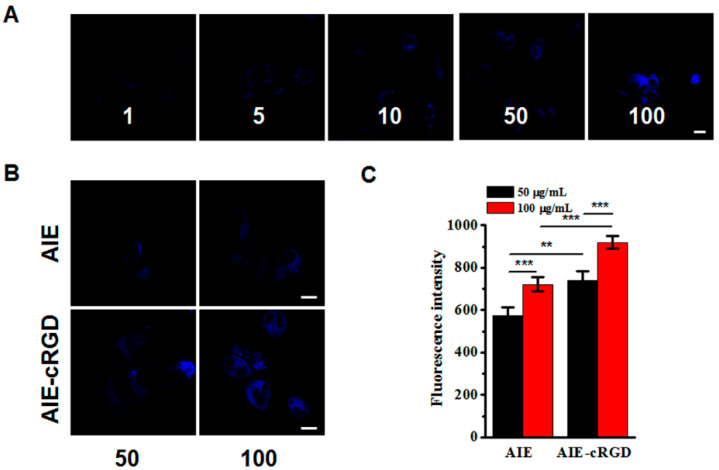
Concentration of probe for labeling the living cells. (**A**) Confocal images of MDA-MB-231 cells after incubation with different concentrations AIE–cRGD (1, 5, 10, 50, 100 μg/mL, in PBS) at 4 °C overnight. (**B**) Confocal images of MDA-MB-231 cells. Incubated with AIE and AIE–cRGD with different concentrations (50 μg/mL, 100 μg/mL μg/mL), respectively. (**C**) MDA-MB-231 cells fluorescence intensity statistical data. The cells were incubated with AIE and AIE–cRGD (50 μg/mL, 100 μg/mL) at 4 °C overnight, respectively. *** *p* < 0.001, ** *p* < 0.01, represented significant differences compared with the control group. Bar = 20 µm.

**Figure 4 cells-11-03478-f004:**
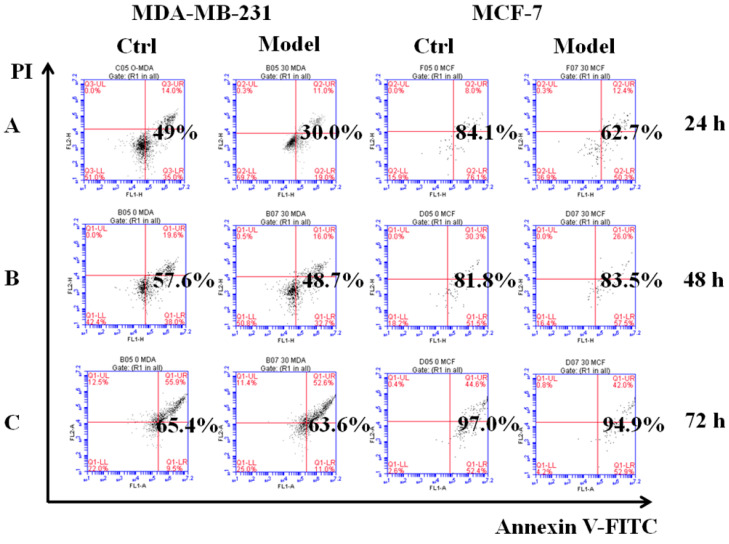
Simulating detached cancer cells with the ability to resist anoikis. The effects of cell apoptosis on MDA-MB-231, MCF-7 cells cultured for 24 h (**A**), 48 h (**B**), 72 h (**C**) in six-well cell culture plates treated with poly-HEMA (30 mg/mL), respectively. As a control, the cells were cultured in low-adhesion plate for 24, 48, 72 h, respectively.

**Figure 5 cells-11-03478-f005:**
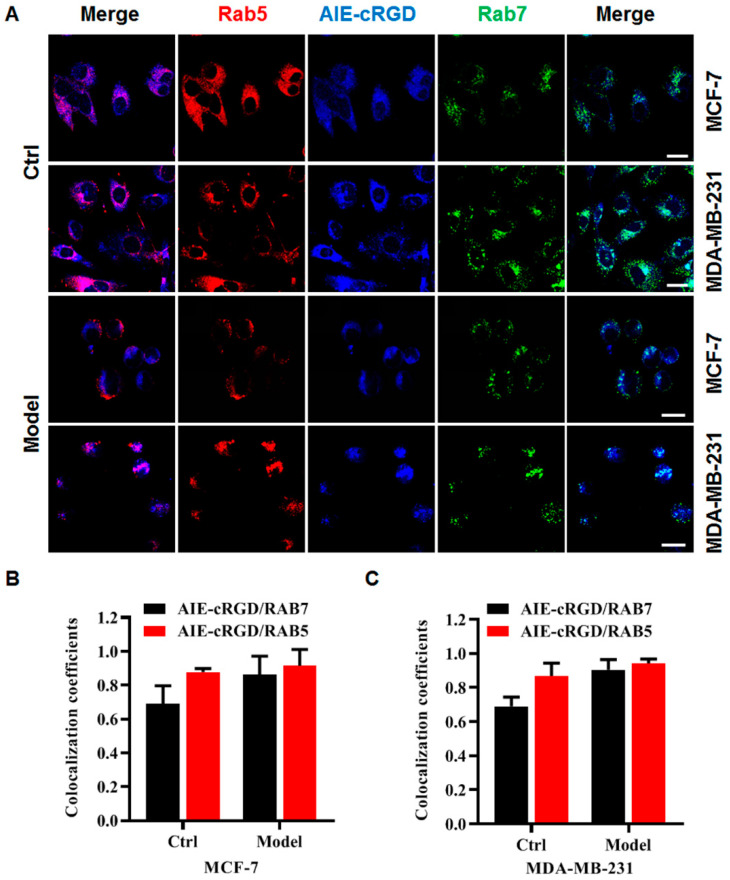
The distribution of integrin clusters in cancer cells. (**A**) Confocal fluorescence microscopy images of MCF-7, MDA-MB-231 cells with different treatments stained with AIE–cRGD, Rab5, Rab7, respectively. Colocalization coefficients of AIE–cRGD with RAB5 or RAB7 in control cells and model cells of MCF-7(**B**), MDA-MB-231(**C**) cells, respectively. Bar = 20 µm.

**Figure 6 cells-11-03478-f006:**
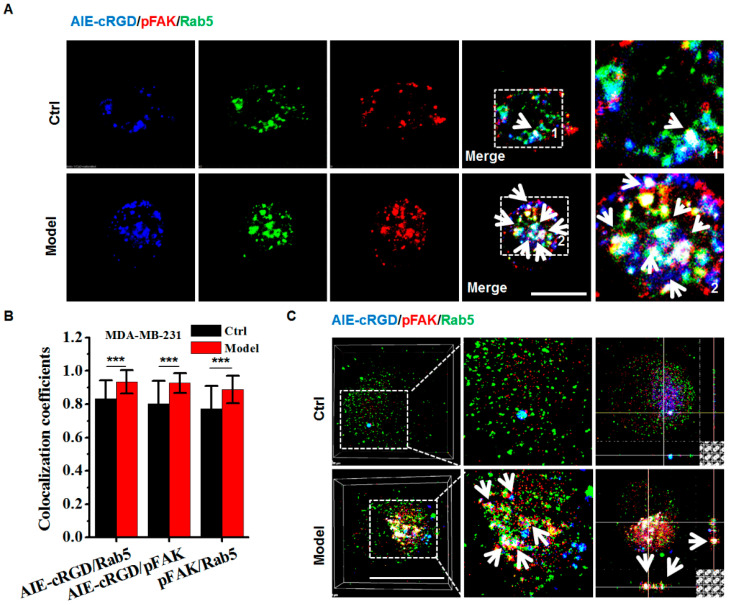
Assaying the resisting anoikis in model cells using the AIE-cRGD. (**A**) Confocal images of MDA-MB-231 cells stained with AIE–cRGD (blue), pFAK (red), and Rab5 (green), respectively. The model cells were cultured in poly-HEMA-coated six-well plates for 24 h, whereas the control was in untreated six-well plates. The white arrows showed the colocation of the AIE-cRGD, pFAK, and Rab5. Bar = 20 µm. (**B**) Colocalization intensity statistic of AIE–cRGD and Rab5, AIE–cRGD and pFAK, and pFAK and Rab5. *** *p* < 0.001 represent significant differences compared with the control group. (**C**) Confocal 3D images of MDA-MB-231 stained with AIE–cRGD, Rab5, and pFAK. The white arrows showed the colocation of the AIE-cRGD, pFAK, and Rab5. Bar = 20 µm.

**Figure 7 cells-11-03478-f007:**
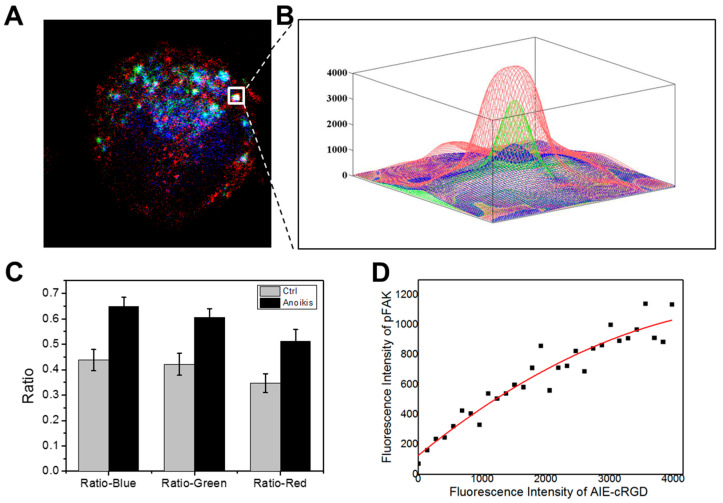
Image analysis. (**A**) Example of three-color high-resolution images collected with the confocal microscope. AIE–cRGD are pseudo-colored in blue, pFAK is pseudo-colored in red, and Rab5 is pseudo-colored in green. (**B**) Close-ups show the fluorescence intensity of the three colors in individual endosomes. (**C**) Comparison of the colocalization of three colors in the proportion of the three colors channels, respectively, between the control group and model cell group. (**D**) Statistics of the relationship of fluorescence intensity between the AIE–cRGD and the pFAK in model cell group. The solid black squares represent pixels in the detection area. The red line in the upper left corner is the picture after fitting the fluorescence point diagram and the linear relation formula.

## Data Availability

Not applicable.

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
