# Peer review of "Fabricated AIE-Based Probe to Detect the Resistance to Anoikis of Cancer Cells Detached from Tumor Tissue"

_cells, 2022, doi:10.3390/cells11213478_

Round 1

Reviewer 1 Report

The manuscript entitled “Fabricated AIE-based probe to detect the resistance of detached cancer cells from tumors tissue to anoikis” by Chang et al. is a relevant study that can involve an increase of knowledge in the field. However, conclusions should be reinforced with new data.

1)      The expression of annexin 5 is interesting but other apoptotic markers such as caspase-3 should be included in the assays.

2)      Authors should clarify the differences observed regarding integrin aggregation-FAK-Rab5/7 clustering during the induction of cell detachment by poly-HEMA coated plates among different cell lines. Figures 5, 6, S3 and S4 show strong differences between CaCo2, HepG2, U85 and McF-7 vs KB, Hela DU145 and MDA-MB-231 in this variable. In this sense, the paragraph 3.6 describes that all cell lines behave similarly, but this assumption does not appear to be fully accomplished in the figures.

Author Response

1) The expression of annexin 5 is interesting but other apoptotic markers such as caspase-3 should be included in the assays.

Answer: Thank you very much. Caspase family is a key element in the process of cell apoptosis, and its activation and overexpression both cause cell apoptosis. However, the method of double staining with fluorescently labeled Annexin V and PI can distinguish cells at different stages of apoptosis (early or late) by flow cytometry, to detect apoptosis. This method may not be able to comprehensively detect apoptosis, but it can achieve the purpose of our research, that is to show the difference in the proportion of apoptosis between each group.

2) Authors should clarify the differences observed regarding integrin aggregation-FAK-Rab5/7 clustering during the induction of cell detachment by poly-HEMA coated plates among different cell lines. Figures 5, 6, S3 and S4 show strong differences between CaCo2, HepG2, U85 and McF-7 vs KB, Hela DU145 and MDA-MB-231 in this variable. In this sense, the paragraph 3.6 describes that all cell lines behave similarly, but this assumption does not appear to be fully accomplished in the figures.

Answer: Thank you so much for pointing it out. According to our data, the cells of each group selected in our experiment showed similar results compared with the control group, that is, an increasement in the rate of colocalization. However, the differences between the groups were the proportion of increase in colocalization rate. These differences can be understood to be caused by differences in cell types. There may be some ambiguity in the expression in the text, which has been modified. Please refer to the red marked part in paragraph 3.6.

Reviewer 2 Report

This may be interesting, but some important points need to be resolved. Notably, a study must provide critical analysis of the data. The authors must clearly define your particular aims and objectives to achieve the above. So, they must develop a critical appraisal of the state of the art. This is an essential element of any article. The primary studies need to address critical scientific questions (both conceptual and methodological). The introduction, written in clear language, covers many relevant issues. Try to rewrite the abstract and conclusions; the way of working is not very well explained, and the procedure is tedious and unsustainable. The interpretation of the results can be improved/reformulated, especially trying to rationalize the marginal differences between the fluorescence intensity of cells incubated with AIE and AIE-cRGD, the same for the colocalization coefficients between the control and the model cells. The authors need to specify if there are significant differences between all these comparisons and try to answer the obvious question, why does the probe show quite a high background? And how do they plan to overcome that? And finally, the authors need to show some critical thinking about their protocol and results; they need to emphasize clearly the defects as well as the advantages of this probe. 

Author Response

This may be interesting, but some important points need to be resolved. Notably, a study must provide critical analysis of the data. The authors must clearly define your particular aims and objectives to achieve the above. So, they must develop a critical appraisal of the state of the art. This is an essential element of any article. The primary studies need to address critical scientific questions (both conceptual and methodological). The introduction, written in clear language, covers many relevant issues. Try to rewrite the abstract and conclusions; the way of working is not very well explained, and the procedure is tedious and unsustainable. The interpretation of the results can be improved/reformulated, especially trying to rationalize the marginal differences between the fluorescence intensity of cells incubated with AIE and AIE-cRGD, the same for the colocalization coefficients between the control and the model cells. The authors need to specify if there are significant differences between all these comparisons and try to answer the obvious question, why does the probe show quite a high background? And how do they plan to overcome that? And finally, the authors need to show some critical thinking about their protocol and results; they need to emphasize clearly the defects as well as the advantages of this probe.

Answer: Thank you so much for your advice. We have made some changes to the abstract and conclusion section. The reason for the high background of the probe is also explained and possible solutions are given, “The water solubility of the probe we designed is not good enough, so the background fluorescence will be high. Subsequent studies can try to improve the water solubility of the probe, so that the aggregated luminescence effect of the probe will be more obvious.” In the conclusion, the existing problems are also emphasized.

Round 2

Reviewer 1 Report

The manuscript entitled “Fabricated AIE-based probe to detect the resistance of detached cancer cells from tumors tissue to anoikis” by Chang et al. is a relevant study that can involve an increase of knowledge in the field. However, conclusions should be reinforced with new data.

1)      The expression of annexin 5 is interesting but other apoptotic markers such as caspase-3 should be included in the assays.

2)      Authors should clarify the differences observed regarding integrin aggregation-FAK-Rab5/7 clustering during the induction of cell detachment by poly-HEMA coated plates among different cell lines. Figures 5, 6, S3 and S4 show strong differences between CaCo2, HepG2, U85 and McF-7 vs KB, Hela DU145 and MDA-MB-231 in this variable. In this sense, the paragraph 3.6 describes that all cell lines behave similarly, but this assumption does not appear to be fully accomplished in the figures. The authors should describes more clearly the figures and data obtained, and avoid any potential confusion and ambiguity.

Author Response

1)The expression of annexin 5 is interesting but other apoptotic markers such as caspase-3 should be included in the assays.

Answer: Thank you very much. Caspase family is a key element in the process of cell apoptosis, and its activation and overexpression both cause cell apoptosis. However, the method of double staining with fluorescently labeled Annexin V and PI can distinguish cells at different stages of apoptosis (early or late) by flow cytometry, to detect apoptosis. This method may not be able to comprehensively detect cell apoptosis, but it is enough in our research, that is to show the difference in the proportion of apoptosis between each group.

2) Authors should clarify the differences observed regarding integrin aggregation-FAK-Rab5/7 clustering during the induction of cell detachment by poly-HEMA coated plates among different cell lines. Figures 5, 6, S3 and S4 show strong differences between CaCo2, HepG2, U85 and McF-7 vs KB, Hela DU145 and MDA-MB-231 in this variable. In this sense, the paragraph 3.6 describes that all cell lines behave similarly, but this assumption does not appear to be fully accomplished in the figures. The authors should describe more clearly the figures and data obtained, and avoid any potential confusion and ambiguity.

Answer: Thank you so much for pointing it out. According to our data, the results in this part described in our manuscript are similar, referring to the higher colocalization coefficient of cells in the model group compared with the control group. The results were similar regardless of the cell line (no matter MDA-MB-231, DU145, Hela, KB, MCF-7, U87, HepG2 or Caco-2). However, the differences in different cell lines were the proportion of increase in colocalization rate. These differences can be understood to be caused by differences in cell types. Integrin is a transmembrane heterodimer, which consists of two non-covalently bound transmembrane subunits, namely α and β subunits. However, the types and quantities of integrins expressed on the surface of different types of cells are different. In this paper, only αvβ3 subtype was studied, and cRGD was selected as the binding peptide in the targeted probe, so that the detection results are limited. The result of this data is that there are differences in signal intensity between cell lines.